# VoxInstruct: Expressive Human Instruction-to-Speech Generation with Unified Multilingual Codec Language Modelling

## ABSTRACT

Recent AIGC systems possess the capability to generate digital multimedia content based on human language instructions, such as text, image and video. However, when it comes to speech, existing methods related to human instruction-to-speech generation exhibit two limitations. Firstly, they require the division of inputs into content prompt (transcript) and description prompt (style and speaker), instead of directly supporting human instruction. This division is less natural in form and does not align with other AIGC models. Secondly, the practice of utilizing an independent description prompt to model speech style, without considering the transcript content, restricts the ability to control speech at a fine-grained level. To address these limitations, we propose *VoxInstruct*, a novel unified multilingual codec language modeling framework that extends traditional text-to-speech tasks into a general human instruction-to-speech task. Our approach enhances the expressiveness of human instruction-guided speech generation and aligns the speech generation paradigm with other modalities. To enable the model to automatically extract the content of synthesized speech from raw text instructions, we introduce speech semantic tokens as an intermediate representation for instruction-to-content guidance. We also incorporate multiple Classifier-Free Guidance (CFG) strategies into our codec language model, which strengthens the generated speech following human instructions. Furthermore, our model architecture and training strategies allow for the simultaneous support of combining speech prompt and descriptive human instruction for expressive speech synthesis, which is a first-of-its-kind attempt.

## CCS CONCEPTS

• **Information systems** → **Multimedia content creation**; • **Human-centred computing** → *Human computer interaction (HCI)*.

## KEYWORDS

Human computer interaction, expressive speech synthesis, codec language model, human instruction, AIGC

### ACM Reference Format:
Anonymous Author(s). 2018. VoxInstruct: Expressive Human Instruction-to-Speech Generation with Unified Multilingual Codec Language Modelling. In *Proceedings of ACM International Conference on Multimedia (MM'24)*. ACM, New York, NY, USA, 10 pages. https://doi.org/XXXXXXX.XXXXXXX

## 1 INTRODUCTION

Human-computer interaction (HCI) aims to enhance user experience and facilitate seamless interactions between humans and computers [3]. With the rapid advancements of deep generative models, recent Artificial Intelligence Generated Content (AIGC) systems can generate digital multimedia content based on human language instructions, such as text [1], image [20], video [24] and audio [17], thereby significantly propelling HCI. Leveraging large-scale training data, these models have achieved remarkable success in text and visual modalities, which can produce high-quality and vivid samples aligned with natural language inputs. However, when it comes to audio, especially speech, there is still significant room for improvement in human instructions-to-speech generation.

In general, speech involves three types of information: linguistic, paralinguistic, and extralinguistic, corresponding to spoken content, prosody/emotion, and speaker/scenario, respectively [18]. Human instructions should be able to describe and control these three aspects within the synthesized speech. Due to the high cost of manually annotating paralinguistic and extralinguistic information in speech, the lack of large-scale datasets with high-quality text-speech pairs constrains the performance of current prompt-based text-to-speech (TTS) models. Besides, existing approaches [9, 12, 15, 23, 31] need to divide inputs into content prompt (transcript) and description prompt (style and speaker), that is less natural in form and does not align with other AIGC models. For example, when performing text-to-image generation, we can use a single natural language prompt to simultaneously describe both the content and style of the image in a flexible way. The practice of using independent description prompts to model speech style embedding, without considering the transcript content, also restricts the ability to control speech at a fine-grained level. Current research on large-scale TTS models [13, 14, 22, 28, 34] primarily focus on using speech prompts for voice cloning. However, relying solely on speech prompts is user-unfriendly and incapable of creating new voices. Furthermore, there is also a gap in current research regarding the simultaneous utilization of both text description prompts and speech prompts for speech generation.

To align the speech generation paradigm with other modalities, we propose VoxIntruct, a new speech generation framework that can directly support human language instructions as inputs, extending the traditional *text-to-speech* task into a general *human instruction-to-speech* task. Specifically, human instructions refer to a combined form freely written by natural language, including both the spoken content and the descriptive information of the speech. Our instruction-to-speech generation model is based on the powerful large language model (LLM) architecture LLaMA [27], and a pre-trained MT5 text encoder [30] is adopted to improve the understanding of instruction context. To enable the model to automatically extract the content of the synthesized speech from raw text instructions, we introduce speech semantic tokens as an

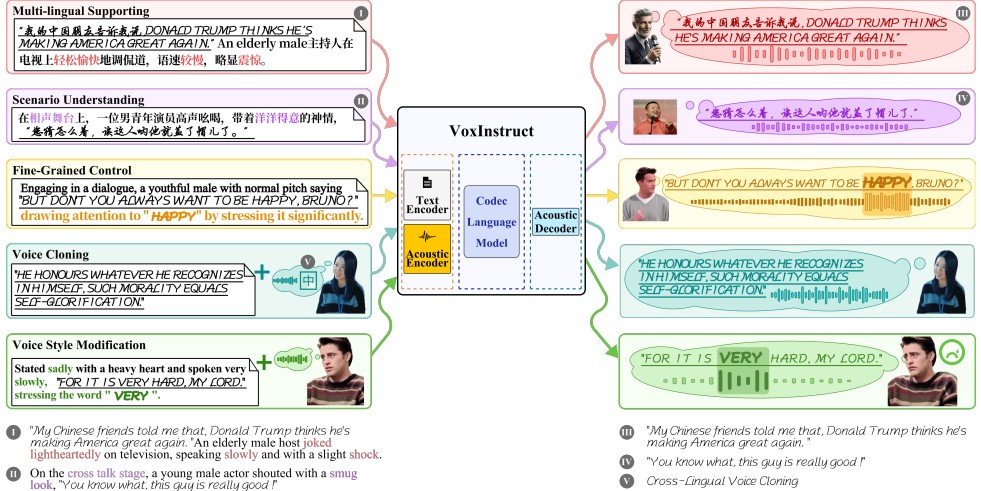

**Figure 1: The capabilities of the proposed expressive human instruction-to-speech generation model.**

intermediate representation for instruction-to-content guidance, eliminating the need for an additional phoneme sequence as the speech transcript, unlike previous approaches. In addition, by incorporating multiple Classifier-Free Guidance (CFG) [21] strategies into our codec language model, we have strengthened the generated speech adhering human instructions. To ehance the model generalization, we adopt the pre-training and fine-tuning paradigm, leading to improvements in terms of expressiveness and naturalness of the synthesized speech. Furthermore, benefiting from the model architecture and training strategies, it is a first-of-its-kind attempt to support inputs that combine speech prompts with descriptive human instruction for expressive speech generation or voice style modification. In particular, when using speech prompts with instructions limited by just spoken content, VoxInstruct operates as a zero-shot voice cloning TTS system, with the performance on par with current state-of-the-art (SOTA) Large-scale zero-shot TTS models for both monolingual and cross-lingual scenarios, demonstrating the comprehensive capabilities of our proposed method.

The contributions of this paper are summarized as follows:

- We present VoxInstruct, the first multilingual codec language modeling framework that extends traditional text-to-speech tasks to general human instruction-to-speech tasks by generating speech directly from human instructions written freely in natural language, replacing previous separate content prompts and description prompts. It significantly improves the expressiveness of synthesized speech and the generalization of prompt-based TTS.
- To strengthen the synthesized speech following human instructions, we introduce speech semantic tokens as an intermediate representation for instruction-to-content guidance, and incorporate multiple Classifier-Free Guidance (CFG) strategies.
- We reveal a successful model architecture and training strategies that support a combination of speech prompts and text description prompts for expressive speech synthesis, which is a first-of-its-kind attempt. It is able to generate speech of competitive quality with current SOTA large TTS models when using only speech prompts.

## 2 RELATED WORKS

### 2.1 Text Prompt-based TTS Methods

In expressive text-to-speech (TTS) synthesis, conventional methods are limited by fixed style labels or reference speech to control the style, which may be inconvenient to users. Therefore, there's growing interest in generating speech from natural language text prompts, with some research already investigating this approach.

PromptTTS [9] utilizes style prompts based on five attributes to direct the stylistic expression of the synthesized voice, and constructs the PromptSpeech dataset containing prompts with style and content information. Additionally, PromptStyle [16] incorporates a reference encoder and aligns text prompt and reference embeddings for cross-speaker style transfer. Emphasizing naturalness and flexibility, InstructTTS [31] enables stylistic speech synthesis using free-form natural language descriptions. Considering that text prompts cannot fully and precisely describe the characteristics of speech, PromptTTS 2 [15] introduces a diffusion-based variation network to address voice variability beyond text prompts, thus tackling the one-to-many issue. PromptSpeaker [33] and PromptTTS++ [23], on the other hand, shift their focus to text description-based speaker generation by incorporating additional speaker information into the text prompts, thereby enhancing control over speaker individuality in speech generation. Each of these models employs a prompt encoder to capture stylistic information from natural language inputs. Meanwhile, Salle [12] treats text-controllable TTS as a language model task, utilizing audio codec codes as an intermediate representation, offering an alternative perspective to text prompt-based TTS systems. And Salle also introduces the textrolspeech dataset, featuring emotion descriptions in prompts.

However, all the above text prompt-based TTS methods input the transcript and description separately. Our model supports more natural prompt inputs by integrating the description and transcript, moving closer to true control via natural language instructions. Additionally, while existing approaches utilize training datasets of limited size and have restricted coverage in terms of domain

(mostly sourced from audiobooks), we have expanded the scale and scope of our data, enabling better and more diverse outcomes.

## 2.2 Large-Scale TTS Models

The remarkable success of large models in text and image generation has indeed spurred significant developments in large-scale text-to-speech (TTS) models. VALL-E [28], for instance, pioneered a codec language modeling approach for TTS by using an audio discrete codec model [8], marking a departure from traditional continuous signal regression methods. It expanded the TTS training data to 60K hours of English speech, leading to substantial advancements in zero-shot voice cloning. VALL-E X [34] extends these capabilities into cross-lingual speech synthesis, further broadening the applicability. Similarly, Spear-TTS [14] regarded TTS as two sequence-to-sequence tasks: from text-to-high-level semantic tokens and semantic tokens to low-level acoustic tokens, employing language models for both stages. Models like VALL-T [7] and RALL-E [29] aimed to improve the stability of these decoder-only LM-based TTS systems by introducing shifting relative position embeddings or chain-of-thought prompting techniques. These models, thanks to their extensive training data and the in-context learning ability of their language model backbones, are capable of producing high-quality and natural speech that closely resembles the speech prompts. Another line of large-scale TTS models leverages non-autoregressive (NAR) modeling, exemplified by systems like NatrualSpeech 2 [22] and Mega-TTS 2 [13]. They also show great zero-shot voice cloning capabilities, and even better robustness than language model-based methods because of explicit duration modeling. However, they typically fall short in achieving the diversity of generated speech compared to AR models. Besides, existing NAR models require extra effort to derive duration alignment from large-scale speech corpora, which can be time-consuming and prone to inaccuracies in noisy speech conditions.

In this paper, we build a human instruction-to-speech generation model based on codec language modeling. Unlike previous TTS codec language models that rely on phoneme sequences, our model directly generates speech from human language instruction. This allows the model to understand unified language instructions that incorporate both spoken content and voice/style description, enabling it to produce expressive speech that adheres closely to the given instructions.

## 3 PROBLEM FORMULATION

Let $x_{ins}$ represents the natural language text of the human instruction that describes the characteristics of voice (speaker's gender, age, speed, pitch), the speaking style (emotion, prosody), the speaking scenario, together with the transcript of spoken content. Our major task is to generate a speech signal $y \in \mathbb{R}^L$ in accordance with $x_{ins}$, where $L$ is the length of samples in $y$. Due to the challenge of directly generating waveforms, it is common practice to first produce an intermediate acoustic representation $A \in \mathbb{R}^{T \times D}$, such as mel-spectrograms or codec, where $T$ is the downsampled length of speech (that is, frame) and $D$ is the feature dimension of each frame, and then utilize an additional vocoder to synthesize the waveform. Hence, the human instruction-to-speech generation process can be briefly defined as $\mathcal{F} : x_{ins} \mapsto A$.

Intuitively, $x_{ins}$ includes the content part $x_{con}$ and the description part $x_{des}$, which represent what is to be said and how it is to be said, respectively.

Conventional TTS task aims to model the transcript-to-speech mapping $\mathcal{F}_{TTS} : x_{con} \mapsto A$. To achieve controllable expressiveness in speech, recent prompt-based TTS works further model the process of $\mathcal{F}_{P-TTS} : (x_{con}, x_{des}) \mapsto A$. However, they require the distinguished inputs of the content prompt and the description prompt, with $x_{des}$ only allowing for coarse control of the overall speech, which is not true instruction-based speech generation.

Unlike them, our proposed speech generation model is designed to directly support human instructions $x_{ins}$ as input, where $x_{ins}$ is a flexible combination of $x_{con}$ and $x_{des}$. For instance, the spoken content $x_{con}$ can be placed before, after, or even inserted at any point within $x_{des}$, and $x_{des}$ can describe the style of either the whole or a portion (such as emphasizing a particular word) of $x_{con}$, much like the structure of novel or article writing. We believe this input format not only facilitates user-friendly instruction-based speech generation but also holds the potential for expansion into a broader and general instruction-based audio generation framework.

In addition, since textual human instructions may be incapable of precisely describing the voice timbre desired by the user, the system should also support speech prompts as an auxiliary optional input. Given a reference speech $\widetilde{y}$ as speech prompt, $\widetilde{A}$ is the intermediate representation of speech prompt encoded from $\widetilde{y}$. In this situation, human instruction and speech prompt complement each other to generate speech, represented as $\mathcal{F}' : (x_{ins}; \widetilde{A}) \mapsto A$. The model takes into account both the detailed voice characteristics in $\widetilde{A}$, and the style and content controls in $x_{ins}$. Specifically, when $x_{ins}$ is limited to contain the spoken content $x_{con}$ only, the model operates as a conventional voice cloning TTS system, synthesizing the given transcript by entirely mimicking the reference speech prompt.

## 4 METHODOLOGY

In this section, we first provide an overview of the human instruction-to-speech generation framework, following which we introduce the core component of this framework - the multilingual codec language modeling based on natural language instruction inputs. Together with a powerful language model architecture LLaMA, speech content guidance with semantic tokens, multiple classifier-free guidance strategies, and pre-training with the fine-tuning paradigm, the proposed system can directly generate high-quality and expressive speech in both English and Mandarin adhering to human language instructions.

## 4.1 Framework Overview

As illustrated in Fig.1, the proposed speech generation framework is made up of a text encoder, an acoustic encoder, an acoustic decoder and a neural codec language model. The detailed architecture of the proposed model is shown in Fig.2. Drawing from the success of other cross-modal generation systems, we utilize a pre-trained text encoder to capture the semantic information of human instruction. To support multilingual instruction inputs, we choose the Multilingual T5 base model (MT5-base) [30][1], and use its pre-trained text encoder with inserting trainable low-rank adaptation (LoRA) adaptors. The

---

[1]https://huggingface.co/google/mt5-base

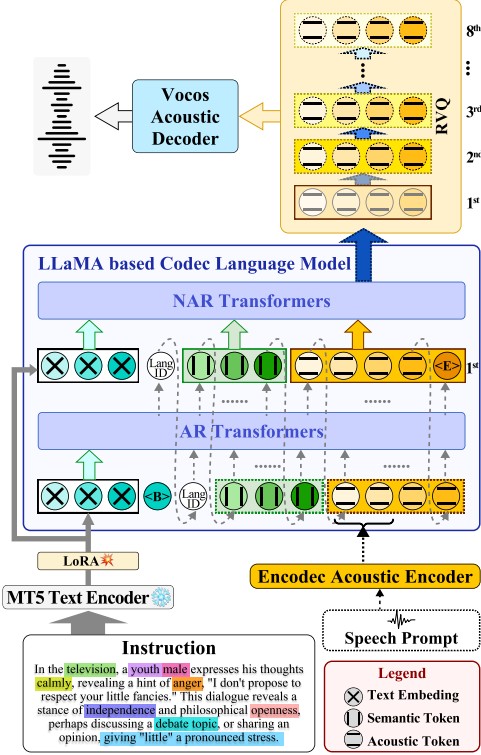

**Figure 2: Model architecture.**

raw text of human instruction $x_{ins}$ is passed to the MT5 encoder to derive the text embedding sequence $E_{ins} = \{e_1, e_2, ..., e_m\}$, where $m$ is the number of subwords after text tokenization. As to the acoustic encoder, the neural codec model Encodec [8][2] is used to extract the discrete acoustic tokens as intermediate representations $A^{T*n}$, where $n$ is the number of residual quantizers of each frame.

Since speech is a type of variable-length sequence data, we employ a codec language modeling approach to model the mapping from instruction text embedding sequence to acoustic tokens, which allows to avoid the need for additional duration prediction. The codec language model takes instruction text embedding $E_{ins}$ and speech prompt $\widetilde{A}$ (if it is provided) as input to produce target acoustic tokens. After generating acoustic tokens, we leverage Vocos [25][3] as the acoustic decoder instead of the original Encodec decoder, as Vocos offers better audio reconstruction quality.

## 4.2 Instruction-to-AT Generation with LLaMA Architecture and ST Guidance

The neural codec language model aims to generate acoustic tokens (AT) based on the text embedding sequence of multilingual human language instructions. Previous LM-based TTS models such as VALL-E mainly adopt a two-stage manner, including autoregressive (AR) and non-autoregressive (NAR) models. The AR model generates the coarse-grained acoustic tokens (the first quantizer) step by step, while the NAR model generates the acoustic details (the rest quantizers) in parallel. Similarly, we also combine AR and

NAR models to ensure both generation quality and inference efficiency, and we leverage a more powerful transformer architecture LLaMA [27] as the model backbone. LLaMA introduces several improvements including pre-normalization, RMSNorm, SwiGLU activation function, and rotary positional embeddings (RoPE) [26], all of which have been proven effective in LLM. Besides, we use fast and memory-efficient flash attention [6] to replace the original attention module in LLaMA.

Unlike text-to-image generation, speech typically requires stricter content alignment. The precise occurrence and the correct order of pronunciation units significantly impact the intelligibility of the generated speech. We found that directly learning the mapping from instruction text embedding to acoustic tokens (AT) is relatively challenging. To enhance the model's understanding of human instructions and generate intelligible speech, we introduce semantic tokens (ST) extracted from speech. These tokens assist the model in discerning the content within $x_{ins}$, eliminating the requirement for supplementary phoneme sequence input. Therefore, our codec language model consists of three stages: instruction-to-ST generation, coarse-grained AT generation, and acoustic details generation, with the first two stages being modeled by the AR model.

*4.2.1 Stage I (AR): instruction-to-ST generation.* To obtain speech semantic tokens, we use the self-supervised representation model HuBERT [11] as well as the $k$-means clustering to discrete HuBERT embeddings[4]. Semantic tokens are expected to provide high-level abstract representations of speech content devoid of prosodic elements (such as duration) or speaking style. Consequently, consecutive duplicate tokens are removed, following the method outlined in [14]. To facilitate the generation of multilingual speech, we prepend a language label $l$ to the semantic token sequence $S$, serving to signify the language information of the speech content. The AR model initially predicts the language label from the instruction text embedding $E_{ins}$, and subsequently produces all semantic tokens, with a $<S_{eos}>$ token indicating the end of ST prediction. The process can be formulated as:

$$P(S|E_{ins}; \theta_{AR}) = P(l|E_{ins}; \theta_{AR}) \prod_{t=1}^{T'} p(S_t|E_{ins}, l, S_{<t}; \theta_{AR}) \quad (1)$$

*4.2.2 Stage II (AR): coarse-grained AT generation.* The first quantizer of acoustic tokens encapsulates essential content, prosody and fundamental timbre information, while also determining the overall speech duration, akin to creating a rough sketch of the speech. We employ the aforementioned AR model to predict such coarse-grained acoustic tokens. The prediction conditions incorporate the instruction text embedding sequence, alongside the language label and the semantic token sequence. This stage can be delineated as follows:

$$P(A_{(:,1)}|E_{ins}; \theta_{AR}) = \prod_{t=1}^{T} P(A_{(t,1)}|E_{ins}, l, S, A_{(<t,1)}; \theta_{AR}) \quad (2)$$

It is unnecessary to distinguish between stage 1 and stage 2 during training, as the input sequence to the AR model is presented in a concatenated form, denoted as $<E_{ins}, l, S, A_{(:,1)}>$. With a causal attention mask in the AR model, all preceding tokens are treated as

[2]https://github.com/facebookresearch/encodec
[3]https://github.com/gemelo-ai/vocos
[4]https://github.com/facebookresearch/fairseq/tree/main/examples/hubert

conditioning elements or prompts (including speech prompts), thus allowing for simultaneous training of both ST and coarse-grained AT generation processes.

*4.2.3 Stage III (NAR): acoustic details generation.* For efficient inference, we utilize a NAR model to generate the rest quantizers of acoustic tokens, which is also based on the LLaMA backbone but omits the casual attention mask. Each layer's quantizer is forecasted using all preceding layers' quantizers as well as the instruction text embedding, the language label, and the semantic tokens. To further improve speech quality, we adopt iterative parallel decoding across each layer, similar to MaskGIT [4] and SoundStorm [2]. In the course of predicting each layer's quantizer, the NAR model performs multiple forward passes, during which it predicts and then retains a portion of the tokens based on their confidence scores. Moreover, to support both instruction-to-speech generation and voice cloning capability, the NAR model is designed to optionally accommodate a speech prompt $\widetilde{A}$. During training, we decided with a certain probability whether to use a prefix segment $\widetilde{A}$ of acoustic tokens as a speech prompt. The entire process can be simply represented as:

$$P(A_{(:,2:n)}|E_{ins}, \widetilde{A}; \theta_{NAR}) = \prod_{i=2}^{n} P(A_{(:,i)}|E_{ins}, l, S, \widetilde{A}_{(:,1:n)}; \theta_{NAR}) \tag{3}$$

## 4.3 Classifier-Free Guidance for Codec Language Model

The success of classifier-free guidance (CFG) in text-to-image generation [10] has demonstrated the effectiveness of combining unconditional generation and conditional generation within diffusion models. Recent advancement in unimodal text generation has further illustrated that CFG can also be used in LLMs [21], improving both coherence and alignment with the given prompt. Motivated by this, we first attempt to introduce CFG into codec language models, to enhance the control over human instruction-to-speech generation.

Specifically, the condition in Equation (1) and (2) are replaced with an empty prompt at a certain probability during AR model training. That is, we mask text embedding sequences when predicting semantic tokens, and we mask text embedding sequences or semantic token sequences when predicting coarse-grained acoustic tokens, both of which are considered forms of unconditional generation. Consequently, during inference, we can sample the *i*-th semantic token in the logits space, combined with unconditional guidance:

$$log\hat{P}(S_t|E_{ins}, l, S_{<t}) = logP(S_t|E_{ins}, l, S_{<t}) \\ + \gamma(logP(S_t|x_{ins}, l, S_{<t}) - logP(S_t|\emptyset, l, S_{<t})) \tag{4}$$

where $\gamma$ is the guidance strength. Besides, when sampling the *i*-th coarse-grained acoustic token, we can utilize two types of CFG at the same time, allowing the generation of AT to focus on different aspects:

$$log\hat{P}(A_{(t,1)}|E_{ins}, l, S, A_{(<t,1)}) = logP(A_{(t,1)}|E_{ins}, l, S, A_{(<t,1)}) \\ + \alpha(logP(A_{(t,1)}|E_{ins}, l, S, A_{(<t,1)}) - logP(A_{(t,1)}|\emptyset, l, S, A_{(<t,1)})) \tag{5}$$

$$log\hat{P}'(A_{(t,1)}|E_{ins}, l, S, A_{(<t,1)}) = log\hat{P}(A_{(t,1)}|E_{ins}, l, S, A_{(<t,1)}) \\ + \beta(log\hat{P}(A_{(t,1)}|E_{ins}, l, S, A_{(<t,1)}) - logP(A_{(t,1)}|E_{ins}, l, \emptyset, A_{(<t,1)})) \tag{6}$$

where $\alpha$ and $\beta$ are the guidance strength corresponding to the human instruction and the semantic tokens. The guidance strength is usually set to be over 1.

Intuitively, enhancing guidance on instructions contributes to better control over the voice characteristics of generated speech, while intensifying guidance on ST helps increase the intelligibility of the speech content, which can be shown in experiments.

## 4.4 Training Strategy

Compared to text-image data, the scale of existing instruction-speech datasets is relatively small. To improve the performance of the proposed speech generation model, we implement a pre-training with fine-tuning paradigm.

In the pre-training stage, we train our model using large-scale public speech datasets that consist only of text transcriptions. The raw transcripts, enclosed in quotation marks, serve as human instructions, which means $x_{ins}$ is limited to the speech content $x_{con}$. This phase ensures that the codec language model exhibits strong text-to-speech synthesis (intelligibility) and zero-shot voice cloning (generalization) capabilities. Subsequently, we fine-tune the model with instruction-speech paired data, endowing it with the ability to understand descriptive information $x_{des}$ in human instructions. The instructions here primarily describe the overall speech attributes in addition to the spoken content. Owing to the relative scarcity of instructions annotated with fine-grained attributes, such as stress marking, we employ a progressive fine-tuning strategy to achieve fine-grained control over speech. Specifically, we further fine-tune the model using a small dataset of fine-grained instructions, thereby equipping the model with the capability for detailed control over speech characteristics.

## 5 EXPERIMENTS

### 5.1 Implementation Details

**Dataset** In line with the scene intention of pre-training and fine-tuning paradigm, we incorporated substantial data varied in annotation granularity, denoted as transcript-only data, instruction data and fine-grained instruction data, as presented in Table 1. Large-scale publicly available speech datasets with transcripts, including Chinese corpus WenetSpeech [32] and English corpus GigaSpeech [5], are firstly involved in pre-training stage. We filtered out samples that are shorter than 3 seconds and those of low quality, resulting in a total of 13.4K hours of speech.

Subsequently, leveraging our internal annotation system, we employed a series of instruction-speech paired datasets with comprehensive and in-depth interpretation of speech expressiveness through diverse natural language instructions. Speech instructions characterise the speech in terms of spoken content, acoustic properties, speaker identity, emotional tone and scenario background, with a subset of fine-grained description towards word emphasis. The detail of the annotation system, encompassing expert classifiers and captioning model, followed by a LLM for instruction rewriting, is elaborated in Appendix. We automatically annotated instructions

**Table 1: Statistics of the Training Data**

| Version | Language | Data Source | #Used Clips | #Duration |
|---|---|---|---|---|
| Transcript-Only | EN | WenetSpeech | 5,746,972 | 6,319h |
| | ZH | GigaSpeech-xl | 5,705,080 | 7,117h |
| Instruction | EN | GigaSpeech-m, LibriTTS-R, TextrolSpeech, In-the-wild Corpus | 1,065,182 | 1,331h |
| | ZH | AISHELL3, Zhvoice, In-the-wild Corpus | 1,233,355 | 1,116h |
| Fine-grained Instruction | EN | LibriTTS-stress | 75,654 | 149h |
| | ZH | AISHELL3-stress | 63,258 | 51h |

on some open-source datasets. Additionally, to enhance generalizability, we collected a considerable corpus of scenario-enriched, in-the-wild audio data from the Internet. This corpus includes a variety of explicit contextual information, ranging from live commerce and news broadcasts to classroom lectures and gaming commentary, equipped the model with stronger generalisation ability on specific scenes. The instruction datasets and the fine-grained instruction datasets contain 2.4K and 200 hours of speech, respectively.

**Training Details** The model training was conducted on 8 NVIDIA A100 GPUs. Initially, the model underwent pre-training for 1M iterations with a batch size of 64, using a gradually decay learning rate starting from $10^{-4}$. A warm-up strategy was employed during the first 10,000 iterations. Following this, the model was fine-tuned on the instruction datasets for 800K iterations with a batch size of 32, and underwent an additional 100K iterations of fine-tuning on the fine-grained instruction dataset.

In terms of model configuration, both the AR model and the NAR model are built on the LLaMA architecture, which includes 12 layers of Transformers with a hidden dimension of 1024 and a feedforward network dimension of 4096. The LoRA adapters inserted within MT5 text encoder have an r value of 16. For the AR model, to facilitate unconditional generation as part of CFG, we mask the entire text embedding sequence or semantic token sequence with a probability of 0.1 during training. For the NAR model, to support the optional input of speech prompts, we set a probability of 0.3 for not using any prefix acoustic segment $\widetilde{A}$. To enable iterative decoding, we employ a cosine schedule to randomly mask a portion of acoustic tokens for the current layer's quantizier.

**Evaluation Metrics** To verify the effectiveness of our proposed speech generation model, we use multiple subjective and objective evaluation metrics. Given the model's capabilities in instruction-to-speech generation and voice cloning, we specifically introduce evaluation metrics focused on these two aspects.

For instruction-to-speech generation, we employ two mean opinion score (MOS) tests to evaluate the quality and controllability of the generated speeches: **MOS-Q** measures the quality of speech, with higher values signifying greater speech quality, naturalness and expressiveness, **MOS-I** measures how well the speech follows the given human instructions, with higher values indicating better control of the speech attributes corresponding to the descriptive instructions. In terms of objective metrics, we perform ASR with Whisper medium model [19][5] on the generated speech and calculate the word error rate (**WER**) with original transcriptions. We also calculate the **accuracy** on several speech attribute factors of the generated speech, with the corresponding classification models.

For voice cloning, we employ **MOS-S** to measure the voice similarity between speech prompts and generated speech. As for objective metrics, speaker embedding cosine similarity (**SECS**) and mel cepstral distortion (**MCD**) are adopted to evaluate the disparity between generated speech and the speech prompt. We employ Resemblyzer[6] to extract the utterance-level speaker embedding for calculating cosine similarity. For all subjective MOS tests, 20 participants take part in the evaluation and rate on a scale from 1 to 5 with 1 point interval.

## 5.2 Compared Methods

We compared our proposed speech generation model VoxInstruct with several systems of text prompt-based TTS and speech prompt-based TTS, respectively. For text prompt-based TTS, we reproduced **PromptTTS** [9] and **Salle** [12] in multi-lingual version, and trained them on our instruction and fine-grained instruction dataset. Specifically, we processed the instruction text prompt to exclude the content part, aligning with their original setting of modeling content and style separately. And we all used Vocos decoder as their vocoder. For speech prompt-based TTS, we select mono-lingual **Vall-E** [28] and cross-lingual **Vall-E X** [34] as baselines. Due to the high cost of reproduction, we directly collected some audio samples from their demo pages[7] for comparison.

## 5.3 Human Instruction-Controlled Speech Generation

To demonstrate the capability of our proposed VoxInstruct in converting human instructions into expressive speech, we first conducted both subjective and objective experiments on an English test set. The test samples were taken from GigaSpeech-s, which were unseen during training. We utilized our instruction annotation pipeline to produce corresponding human instructions for these samples. As ground truth speech is available, we were able to compute the MCD and SECS with ground-truth as references in this part. The results are presented in Table 2. It is evident that VoxInstruct achieved the highest MOS-Q of 4.22 and MOS-I of 3.76, outperforming the two baseline models significantly. The speech quality of our reproduced PromptTTS is relatively low, which may be attributed to its Transformer-decoder architecture and the use of MSE loss, as mentioned in [23]. This seriously affects its subjective evaluation results and the WER value. For objective evaluations, VoxInstruct also secured the best average classification accuracy of speech attribute factors with the closest similarity to ground-truth speech and obtained a considerable WER of 2.5 which is in line with other state-of-the-art TTS systems. This indicates that VoxInstruct

---

[5]https://github.com/openai/whisper

[6]https://github.com/resemble-ai/Resemblyzer
[7]https://www.microsoft.com/en-us/research/project/vall-e-x/

**Table 2: The experimental results of human instruction-to-speech generation on the English test set**

| Model | MOS-Q | MOS-I | Accuracy on Speech Attribute Factors | | | | | | | WER↓ | MCD↓ | SECS (GT)↑ |
| | | | Mean | Gender | Age | Pitch | Energy | Speed | Emotion | | | |
|---|---|---|---|---|---|---|---|---|---|---|---|---|
| Ground Truth | - | - | 80.12 | 100.00 | 55.06 | 81.01 | 62.03 | 82.59 | 100.00 | 2.7 | - | - |
| PromptTTS | 1.82 | 2.07 | 66.93 | 78.80 | 58.23 | 60.76 | 60.13 | 70.25 | 73.42 | 11.6 | 16.161 | 0.577 |
| Salle | 3.67 | 3.18 | 65.45 | 85.13 | **59.49** | 57.59 | 60.44 | 63.92 | 66.14 | 7.2 | 12.768 | 0.595 |
| VoxInstruct w/o pre-training | 4.11 | 3.66 | 73.95 | 94.62 | 54.43 | **77.53** | **61.39** | 77.22 | 78.48 | 3.3 | 16.273 | 0.622 |
| VoxInstruct | **4.22** | **3.76** | **74.89** | **95.57** | 57.28 | **77.53** | 59.81 | **78.16** | **81.01** | **2.5** | **11.864** | **0.641** |

**Table 3: The experimental results of speech generation from instructions based on randomly sampling speech attributes and LLM-aided generation in Chinese**

| Model | MOS-Q | MOS-I | Accuracy on Speech Attribute Factors | | | | | | |
| | | | Mean | Gender | Age | Pitch | Energy | Speed | Emotion |
|---|---|---|---|---|---|---|---|---|---|
| PromptTTS | 2.36 | 2.17 | 65.38 | 85.61 | 60.00 | 54.74 | 41.40 | 70.53 | 80.00 |
| Salle | 2.77 | 2.68 | 61.22 | 87.37 | 58.95 | 49.12 | 44.56 | 56.14 | 71.23 |
| VoxInstruct w/o pre-training | 3.56 | 3.72 | 63.75 | 94.74 | 56.14 | 54.39 | 44.21 | 66.32 | 66.67 |
| VoxInstruct | **4.01** | **3.83** | 61.4 | 89.47 | 57.89 | 54.04 | 43.51 | 57.54 | 65.96 |

possesses the ability to understand unified human instructions $x_{ins}$, capable of recognizing descriptions of voice characteristics and accurate spoken content within the instructions and generating expressive speech that is consistent with the given instructions. Moreover, it can be observed that incorporating a pre-training phase results in a slight improvement in speech attribute control and a more pronounced enhancement in intelligibility, which is intuitively expected.

In addition, we further conducted experiments in Chinese. Unlike the English test, Chinese instructions were generated by first randomly sampling speech attributes and then leveraging an LLM for rewriting. The results are outlined in Table 3. Although VoxInstruct's performance in objective accuracy metrics is comparable to, or slightly inferior to, the baseline models, it excels significantly in the subjective metrics of MOS-Q and MOS-I, with 4.01 and 3.83, respectively. This demonstrates that our model also performs well in understanding Chinese instructions and generating speech. Furthermore, our findings reveal that VoxInstruct can inherently comprehend mixed-language instructions and produce code-switched speech directly, eliminating the need for any grapheme-to-phoneme (G2P) conversion.

**Table 4: The recall accuracy of speech stress detection**

| Model | Acc_word | Acc_sentence |
|---|---|---|
| PromptTTS | 76.46 | 65.59 |
| Salle | 81.75 | 71.96 |
| VoxInstruct | **88.29** | **87.17** |

## 5.4 Speech Stress Control through Fine-Grained Human Instructions

To demonstrate that our unified instruction-based speech generation approach has superior fine-grained control over speech, we fine-tuned all these models on a fine-grained instruction dataset and evaluated the performance by using an internal stress detection model. 200 instructions containing detailed emphasis information were used to synthesize test samples.

The accuracy of correctly detecting stressed words among all words (**Acc_word**) in synthesized speech, as well as the accuracy of identifying the correct stressed word among all sentences (**Acc_sentence**), are displayed in Table 4. It is shown that the method used by PromptTTS, which focuses on the mapping between text prompts and global speech style embeddings, encounters difficulties in achieving fine-grained control. Conversely, our approach, which utilizes unified instruction prompts as input, shows superior fine-grained control capabilities compared to Salle's method, which models content and style prompts separately.

## 5.5 Voice Cloning Ability Based on Speech Prompt

In this section, we compare VoxInstruct with the zero-shot TTS model VALL-E and Vall-E X, which respectively focus on monolingual and cross-lingual scenarios. The results are depicted in Table 5. This comparison reflects that our model achieves performance comparable to current leading zero-shot voice cloning TTS models. Despite being fine-tuned on instruction datasets, it retains its power capability for mimicking the voice from a speech prompt. Additionally, our model significantly outperforms VALL-E in terms of naturalness and speech quality. This improvement is attributed to the Vocos Decoder and the enriched semantic information provided by the pre-trained MT5 Text Encoder.

**Table 5: The experimental results of voice cloning**

| Model | MCD↓ | SECS↑ | MOS-Q | MOS-S |
|---|---|---|---|---|
| Vall-E | 7.042 | 0.839 | 3.48 | 3.99 |
| VoxInstruct (*mono-lingual*) | 7.503 | 0.824 | 4.06 | 4.01 |
| Vall-E X | - | 0.811 | 4.01 | 3.85 |
| VoxInstruct (*cross-lingual*) | - | 0.816 | 3.68 | 3.86 |

## 5.6 Ablation Studies

To demonstrate the effectiveness of our proposed method for human instruction-to-speech generation, we conducted ablation studies about the specific designs. The base setting for the ablation studies

is configured as VoxInstruct with all CFG values set to 1.0 during inference, and without the pre-training stage to conserve training costs.

For our proposed multiple CFG strategies, we individually set the corresponding CFG values to 2.0 (while maintaining others at 1.0) to explore the impact of enhancing condition guidance on the predictions of different components. We found that enhancing the instruction guidance for semantic tokens and coarse-grained acoustic tokens can improve the accuracy of speech attribute control, with a more significant impact on acoustic tokens. Additionally, enhancing the semantic token guidance for coarse-grained acoustic tokens generation can improve the intelligibility of speech. We experimented with removing semantic token guidance from the codec language model and found that it led to a significant increase in WER. This indicates that incorporating ST sequence helps the model learn and understand the correct spoken content within the instructions.

Table 6: Ablation studies on the English test set

| Model | WER↓ | MCD↓ | SECS↑ | Acc↑ |
|---|---|---|---|---|
| **VoxInstruct (w/o CFG) #1** | 3.0 | 11.861 | 0.609 | 68.51 |
| #1 + CFG of Instruction on ST | 3.5 | 12.441 | 0.607 | 69.41 |
| #1 + CFG of Instruction on AT | 2.7 | 11.692 | 0.615 | 71.41 |
| #1 + CFG of ST on AT | 2.5 | 11.704 | 0.614 | 68.40 |
| #1 - ST guidance | 26.2 | 14.146 | 0.599 | 62.34 |

## 5.7 Case Study

To further explore the capabilities of our proposed VoxInstruct in human instruction-to-speech generation, four case studies are presented. As illustrated in Fig.3, the mel-spectrograms, pitch, and energy contours of speech generated according to human language instructions are depicted.

The first two sub-figures show the controllability of VoxInstruct in generating speech solely from instructions. In Fig.3 (a) and (b), the content of the speech is the same, but the descriptive information differs. The pitch curve rises in Fig.3 (a), corresponding to a happy emotion, and the speech duration is shorter, matching the description "quickly". In Fig.3 (b), the instruction denotes to emphasize the word "always", which is reflected in a higher energy level in the corresponding part of the mel-spectrogram.

In addition, the last two sub-figures show VoxInstruct's capability to achieve voice style modification by using human instructions with speech prompts. The speech prompt used in Fig.3 (c) and (d) is from a male speaker, p254 in the VCTK corpus, which is in a neutral emotion. It can be observed that when different instructions are used, the model can synthesize speech with corresponding global and local styles. For instance, a long pause matching "heavy heart" and "very slowly" in Fig.3 (c), while the word "yielding" is stressed in Fig.3 (d). The SECS values all exceed 0.78, demonstrating that our model effectively maintains timbre consistency while modifying the style, which is a crucial aspect.

## 6 CONCLUSION

In this paper, we propose *VoxInstruct*, a novel unified multilingual codec language modeling framework that extends traditional text-to-speech tasks into a general human instruction-to-speech task.

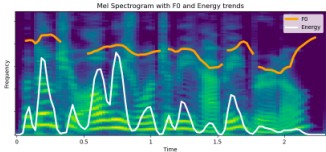

(a) **A happy old man with low pitch and high energy, speaking quickly, happily recounts his recent activities:** "*But don't you always want to be happy, Bruno?*"

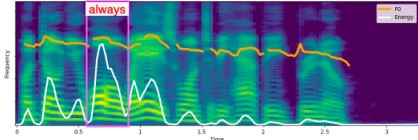

(b) **Engaging in a dialogue, a youthful male with normal pitch saying** "*But don't you always want to be happy, Bruno?*", **drawing attention to** "*always*" **by stressing it significantly.**

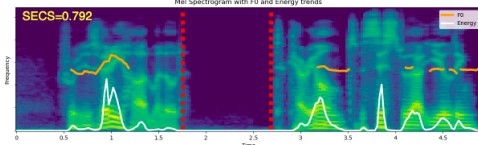

(c) **Stated sadly with a heavy heart and spoken very slowly:** "*For it is very hard, my LORD. To carry on, to persist without yielding.*"[with speech prompt]

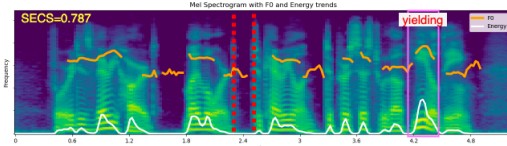

(d) **In the television series, a general said in a calm tone,** "*For it is very hard, my LORD. To carry on, to persist without yielding*", **emphasizing the word** "*yielding*".[with speech prompt]

Figure 3: Mel-spectrograms, pitch, and energy contours of speech generated according to human language instructions for four test cases are depicted. Each subplot is annotated with its respective instruction input. In cases (a) and (b), only the instruction text is provided, whereas cases (c) and (d) also include a speech prompt. The speaker embedding cosine similarity (SECS) between these cases and the speech prompt is displayed in the top left corner.

we introduce speech semantic token as instruction-to-spoken content guidance, multiple Classifier-Free Guidance (CFG) strategies, and pre-training with fine-tuning stage. Our approach enhances the expressiveness of human instruction-guided speech generation and aligns the speech generation paradigm with other modalities. Furthermore, our model architecture and training strategies allow for the simultaneous support of combining speech prompt and descriptive human instruction for expressive speech synthesis, which is a first-of-its-kind attempt.

**ACKS** This work was supported by XXXXXX.

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

# A APPENDIX

