# OpenReview forum: "VoxInstruct: Expressive Human Instruction-to-Speech Generation with Unified Multilingual Codec Language Modelling"
_acmmm.org/ACMMM/2024/Conference — MM2024 Oral_

### Official Review · Reviewer_Mvsa · 2024-05-23

**Rating:** 4
**Confidence:** 3

**Summary:**

This paper presents VoxInstruct, a novel multilingual codec language model designed to generate expressive speech directly from human instructions. Unlike existing methods that require separate content and description prompts, VoxInstruct uses human instructions written in natural language, enhancing the expressiveness and naturalness of synthesized speech. The model incorporates speech semantic tokens and multiple Classifier-Free Guidance (CFG) strategies to improve speech attribute control and intelligibility. The approach is validated through extensive experiments and case studies in both English and Chinese, demonstrating its effectiveness and versatility.

**Strengths:**

The paper demonstrates technical rigor by introducing multiple CFG strategies and speech semantic tokens, showing significant improvements in speech attribute control and intelligibility.

The model is evaluated extensively through quantitative metrics (e.g., WER, MCD, SECS) and qualitative case studies in both English and Chinese. The results show that VoxInstruct performs well across different languages and scenarios, including zero-shot voice cloning.

**Limitations:**

The paper does not sufficiently address potential ethical implications, such as biases introduced by visual data or the impact of automating translation in low-resource languages. A discussion on these aspects would be valuable for understanding the broader impact of the technology.

This approach is quite straightforward and has already been widely applied, so its novelty is not particularly strong.

**Suitability:**

2

---

### Official Review · Reviewer_PyCo · 2024-05-26

**Rating:** 6
**Confidence:** 3

**Summary:**

In this paper, the authors introduce a novel unified multilingual codec language modeling framework VoxInstruct. It differs from previous framework by generating speech directly from human instructions written in natural language, replacing the previous separate content and descriptive prompts. The technique innovation includes instruction-to-spoken content guidance, CFG strategies and pre-training with fine-tuning training stage. The experimental results demonstrate the effectiveness of this framework and its outstanding capabilities in voice cloning and voice style modification.

**Strengths:**

- This paper is very well written and the illustrations is detailed and informative enough.
- The motivation to align the speech generation with other modalities by using human instructions is novel and instereting. And the motivation is supported well in this paper.
- The datasets is quite large and diverse and the experiments including ablation is comprehensive and deep, which show the effectiveness and robustness of the VoxInstruct model.

**Limitations:**

I have some doubts about the training details. It seems that the VoxInstruct model employs three-stage training: pre-training, fine-tuning with instruction data and then fine-tuning with fine-grained instruction data. What about the other baseline methods in terms of the training strategies and can they guarantee the fairness?

**Suitability:**

3

---

### Official Review · Reviewer_Z8gM · 2024-05-27

**Rating:** 4
**Confidence:** 2

**Summary:**

This paper proposes VoxInstruct, a novel unified multilingual codec langauge modeling framework to extend traditional TTS tasks to general human instruction-to-speech generation task. Different from conventional prompt-to-TTS methods that input the transcript and description separately, the proposed model lets the prompt inputs be more natural, by integrating the description and transcript. In detail, authors adipt speech semantic tokens as an intermediate representation for instruction-to-content guideline, and especially work on some classifier-free guidance (CFG) strategies, to show that the proposed architecture and training strategies support a combination of speech and text description prompts for expressive speech synthesis.

**Strengths:**

- Presents novel methods of 1) Instruction-to-AT generation with Llama architecture and ST guidance, and 2) classifier-free guidance for codec language model, along with the thorough formulation that helps the comprehension of the whole architecture.
- The performance of the proposed method is compared with solid and replicated baslines, along with various ablation studies including the speech stress control and voice cloning ability, showing detailed examples with the case study.

**Limitations:**

- It is still not clear how 'multilingual' can be considered crucial in the total system, given that role of both languages seem to differ in the generation process -- it seems to be rather 'code-switching'?
- Acronym used without full reference (e.g,. ln.8 AIGC)

**Suitability:**

3

---

### Meta-Review · Area_Chair_m6xa · 2024-07-01

**Recommendation:** Accept (Oral)
**Confidence:** 4

**Metareview:**

The paper proposes a human instruction to speech generation model. The reviewers mention (+) novelty in the way it can directly work with instruction to speech without need for distangling the content/descriptive prompts, (+) the performance being good and there being various ablation studies, showing robustness of the method. There are some minor concerns on (-) ethical implications such as when adapting it for low resource languages and (-) general multilinguality of the model.

The initial reviews were rather positive with two borderline accepts and one strong accept. Both borderline accepts upgraded their reviews to weak accepts after the rebuttal. All reviewers confirmed they think this paper should be accepted. One reviewer argued strongly for an oral presentation in the rebuttal discussion.

Due to this, I believe this paper should definitively be accepted. Because of the high ratings I would also judge for an oral presentation.